# Efficacy and Immune Response Elicited by Gold Nanoparticle- Based Nanovaccines against Infectious Diseases

**DOI:** 10.3390/vaccines10040505

**Published:** 2022-03-24

**Authors:** Anirban Sengupta, Mohammad Azharuddin, Noha Al-Otaibi, Jorma Hinkula

**Affiliations:** 1Division of Molecular Medicine and Virology, Department of Biomedical and Clinical Sciences, Linköping University, 58185 Linkoping, Sweden; anirban.sengupta@liu.se (A.S.); mazharuddin@cdtltd.co.uk (M.A.); 2King Abdulaziz City for Science and Technology (KACST), Riyadh 11442, Saudi Arabia; naalotaibi@kacst.edu.sa

**Keywords:** gold nanoparticle, nanovaccine, nanoparticle immunology, nanovaccine immunity, GNP

## Abstract

The use of nanoparticles for developing vaccines has become a routine process for researchers and pharmaceutical companies. Gold nanoparticles (GNPs) are chemical inert, have low toxicity, and are easy to modify and functionalize, making them an attractive choice for nanovaccine development. GNPs are modified for diagnostics and detection of many pathogens. The biocompatibility and biodistribution properties of GNPs render them ideal for use in clinical settings. They have excellent immune modulatory and adjuvant properties. They have been used as the antigen carrier for the delivery system to a targeted site. Tagging them with antibodies can direct the drug or antigen-carrying GNPs to specific tissues or cells. The physicochemical properties of the GNP, together with its dynamic immune response based on its size, shape, surface charge, and optical properties, make it a suitable candidate for vaccine development. The clear outcome of modulating dendritic cells, T and B lymphocytes, which trigger cytokine release in the host, indicates GNPs’ efficiency in combating pathogens. The high titer of IgG and IgA antibody subtypes and their enhanced capacity to neutralize pathogens are reported in multiple studies on GNP-based vaccine development. The major focus of this review is to illustrate the role of GNPs in developing nanovaccines against multiple infectious agents, ranging from viruses to bacteria and parasites. Although the use of GNPs has its shortcomings and a low but detectable level of toxicity, their benefits warrant investing more thought and energy into the development of novel vaccine strategies.

## 1. Introduction

The development of vaccines and immunization programs against various kinds of diseases ranging from infections to cancer significantly progressed in the past few decades. One of the most important factors contributing to this growth is the advancement of nanotechnology. The use of nanoparticles in the development of vaccines is a major landmark step. One key step for the vaccine development process is the use of an optimal carrier or delivery system that can influence a potent immune response. The use of different types of nanoparticles and their roles in influencing the immune system has been studied in different disease models. Nanocarriers can be used as adjuvants in the preparation of new-age vaccines. The size, shape, route of administration, and antigen tagging mechanism on the nanoparticles are all critical in this [1,2]. In the past few years, major progress has been achieved in characterizing nanoparticle-based immunogenicity, immunotoxicity, the nature of immune suppression, and immunomodulation [3,4,5].

Among different nanoparticles already tried and used for the successful development of nanovaccines, the gold nanoparticle (GNP) is noteworthy. The chemical synthesis process of antigen tagging on the GNP and its formulations is easy, making it a suitable candidate in the nanovaccine manufacturing process [6]. Metallic nanoparticles such as GNPs provide a higher binding affinity, special electronic structures, plasmon excitation, and large surface energies owing to their higher surface area to volume ratio [7]. This also enables GNPs to interact with different functional groups or ligands with high affinity [8]. Due to its inherent magnetic and optical properties, colloidal gold has already been used in the treatment of a wide variety of diseases with a minimum level of cytotoxicity.

Multifunctional GNPs have been used by conjugating them with FDA-approved antimicrobial drugs and antibiotics in many studies [9,10,11,12,13,14,15]. GNPs coated with antigenic peptides have also been used to synthesize antibodies specific to the pathogens [16,17,18,19,20,21,22,23,24,25,26,27,28,29,30,31,32]. The GNP-based drug or antigen delivery system is more competent for its controlled release to the target site [33]. GNP nanoformulations can be used by tagging specific antibodies or molecules to their surface. This enables efficient targeting to the particular cell types, leading to a site-specific immune response profile and less off-target distribution [34]. The GNP has by itself excellent adjuvant properties to boost the immune system of the host. The variations in size, shape, charge, and surface functionalization are all crucial in eliciting varied immune responses upon administration [35]. Some of these properties are highlighted in Figure 1.

One key concern regarding the use of any foreign chemical such as gold in any form of treatment intervention is its possible harmful side effects. However, the administration of gold as an adjuvant or in a nanovaccine formulation has a high safety profile with few side effects [36,37]. Like any other nanomaterial, GNPs also have certain limitations that we discuss later in this review. However, the advantages of using them for new-age vaccine development are showing promising results and far surpassing the concern.

Although GNPs are widely used in the development of vaccines against multiple cancers, in this review, we focus on their use in vaccination against infectious agents. We discuss the characteristic features of gold nanoparticles that make them advantageous to use in vaccine development, including their shape, size, and generated immune response (Figure 2). We specifically elaborate in separate sections about the use of gold nanovaccines in different types of immune cells and infections: bacterial, viral, and parasitic. We highlight the advancements made in the use of gold nanoparticles in the vaccine development process.

## 2. GNP Characteristics and Features Make It Indispensable in Vaccine Development Research

GNPs are often used to develop a potent antigen carrier system for immunization [1,2,33,34,35,36,37]. They are easy to prepare and have special physicochemical properties with very little toxicity [38]. Multiple variables in the shape, size, geometry, and surface modifications influence GNP function. The stability of the GNP and its flexibility helps in manufacturing GNPs with variable core diameter, size, and shape. Conversion of the electromagnetic radiation to heat by this noble metal can be exploited for therapeutic and targeting purposes [39,40,41,42]. However, there is no relevant systemic study for an optimal and standard GNP system for all applications [41,42].

Precision in the nanocarrier delivery and penetration to the site of interest or the immune cells is a critical component. This facilitates the induction of the immune response genes, antigen processing, cytokine production, antibody secretion, and T cell stimulation for effective therapy or vaccine efficacy [43,44,45,46]. GNPs have unique size and surface area properties. They can penetrate blood vessels and tissue barriers and can deliver to targeted sites due to their high uptake efficiency [37,47,48].

GNPs are efficient in delivering antigens into the major antigen-presenting cells such as dendritic cells, facilitating the downstream immune response, cross-presentation, and CD8+ cytotoxic T cell response (Figure 2A,B) [49]. Along with passive targeting by varying the size and shape of the GNPs to make them more prone to internalization by the individual cell types, active targeting can also be achieved via surface modifications and functionalization. For example, using GNPs coated with antibodies for DEC205, CD40, CD11c, or mannose can be selectively uptaken by dendritic cells by the process of receptor-mediated endocytosis [50,51,52]. For targeting them to macrophages, CD44, folates, and lectins are used [53,54,55]. Thus, loading GNPs with immune target antibodies leads to the activation and stimulation of the specific immune cells.

GNPs are biocompatible and inert. They are easily functionalized with a wide range of peptides and molecules and are also very stable [2,56,57,58,59]. GNPs can be packaged inside virus-like particles (VLP) using the expression of structural genes of the virus and can be used in the vaccine development process [60]. GNPs can be conjugated with the polysaccharide or protein linkers before their antigenic functionalization. The immunomodulating capacity of gold glyconanoparticles is well known. In many vaccine development programs, the GNP is used as an adjuvant to stimulate the immune response [61,62]. Hence, all these features make the GNP a favored choice in biomedical applications for vaccination, drug delivery, and tracking (Figure 1).

The GNP shape, size, charge, and conjugated materials all influence organ accumulation and blood clearance [63]. Progress has been made to optimize GNP pharmacokinetics by increasing the half-life time of its circulation and its physical size and by reducing the mononuclear phagocytosis system (MPS)-based clearance [64]. Polyethylene glycol (PEG)-mediated surface modification of GNPs is commonly used to decrease MPS activity and increase their circulatory half-life [65]. Using 15 nm GNPs provides a better half-life than 100 nm GNPs, while GNPs smaller than 6 nm are rapidly filtered out by the kidneys [66,67]. Protein crown formation on the GNP after its entry into circulation and opsonization facilitate its recognition by MPS of liver, spleen, and marrow, leading to its higher accumulation in these organs [68]. This crown formation also has a crucial impact on biodistribution, as it masks the original functionalization of the GNP [69,70].

## 3. Shape and Size of GNP Influence Its Impact on the Immune System

The size of the GNP, along with its shape, influences the immune system differently. This shape and size dependency of the adjuvant activity of GNPs is used to polarize the immune response in different scenarios to deliver the best outcome [37,38,39,40,41,42].

Rod-shaped GNP-treated bone marrow dendritic cells (BMDCs) produce high levels of IL1b and IL18, whereas cube- and spherical-shaped GNPs result in the production of high levels of pro-inflammatory cytokines such as tumor necrosis factor-alpha (TNFa), interleukin (IL)-6, IL17, and granulocyte-macrophage colony-stimulating factor (GM-CSF) [71]. Niikura et al. [72] studied the varieties of GNP shapes: spherical, rod, and cube. This group found that the ratio between the total surface area per single nanoparticle volume is critical for antibody response and TNFa production. They found that larger-sized (40 nm) spherical GNPs are more efficient in producing IL6, IL12, and GM-CSF than the smaller-sized or differently shaped GNPs [72]. GNP functionalization by chemical modification, addition or removal of functional group, antigen coating, etc., influences surface charge and hydrophobicity, leading to alterations in the level of immune response thus generated (Figure 2) [73,74].

Chen et al. studied the impact of GNP sizes spanning from 2 to 50 nm and found that those between 8 and 12 nm are mostly drained nanoparticles [18]. GNPs between 14 and 20 nm are reported to be better uptaken by the cells. An increase in the diameter leads to more toxicity. Receptor-mediated endocytosis may be the probable mechanism by which GNPs enter the cells [50,51,52,75]. The diameter of a GNP is also correlated with its localization inside the cell. A tiny GNP with a diameter of around 2.4 nm can be localized inside the nucleus, whereas a particle size of around 5.5 to 8.2 nm is found mostly in the cytoplasm. The nanoparticles with a higher diameter above 18–20 nm do not generally enter the cells [39,40,41,42,76].

## 4. Effect of GNPs on Dendritic Cells, Macrophages, and Natural Killer Cells

### 4.1. Dendritic Cells

The effect of GNPs on DCs is critical as it can activate the branch of the adaptive immune system. GNPs surface-tagged with DC-targeting molecules results in the better induction and polarization of immune response (Figure 2A,B) [77]. Although many studies [78,79] suggest the possible cytotoxicity, phenotype alteration, cytokine production, and activation by GNPs targeting DCs, the intricate details of such interaction, stimulation, or suppression of the immune system are yet to be divulged. In one study, DCs loaded with a GNP-conjugated *Listeria* antigen were adoptively transferred to naïve animals, leading to the induction of natural killer cells, CD8 + T cells, and better Th1 response and vaccine efficacy than any other traditional immunization methods [24].

DC from bone marrow, when stimulated by GNP, starts producing IL6, TNF-α, and interferon gamma (IFN-γ) [80]. GNP can induce the extracellular traps for the neutrophils, leading to immune system triggering via DNA receptors such as Toll-like receptor 9 (TLR9) [81]. GNPs coated with polyethylene glycol (PEG) or polyvinyl alcohol (PVA) or both increased their interaction with monocyte-derived dendritic cells (MDDCs). Although PEG coating restricts GNP uptake, it enhances the TNFa synthesis. PVA- or PEG+PVA-coated GNPs have a higher rate of uptaking with IL1b synthesis, although both types of coating do not influence the immunological characteristics, phenotypes, or activation of MDDCs [82].

### 4.2. Macrophages

The polarization and function of macrophages are reported to play a key role in different disease conditions [83]. GNPs are reported to promote crosstalk between the macrophages and other cells for tissue regeneration [84] and also suppress pro-inflammatory cytokine release from the macrophages. Similar suppression of immune response is reported in the lipopolysaccharide (LPS)-treated splenocytes in the presence of GNPs [85,86].

GNPs are reported to have oxygen radical scavenging properties in the mouse macrophages as they reduce the reactive oxygen species (ROS) of GNPs in a dose-dependent manner in the presence of LPS treatment [24]. The reduction in the pro-inflammatory cytokines, including IL17 and TNFa, is also noteworthy in the same set of experiments [87].

### 4.3. Natural Killer Cells

Natural killer cells (NK) sourced from the lymphoid progenitor lineage play a crucial role in immune surveillance in the circulation. They release granzymes and perforins to induce infected cell lysis. GNPs have been researched to target NK cells by using the mechanism of NK cell-mediated antibody-dependent cellular cytotoxicity (ADCC). This is the use and targeting of the NK cell receptors by antibody-tagged GNPs. This helps in the delivery and activation of NK cells [87].

PEGylated polyamidoamine dendrimers entrapped within GNPs to transfect human ferritin heavy chain in the NK cells constitute a novel immunotherapy method [88]. A modified version of GNPs has an anti-inflammatory response while being used to treat NK cells in vitro, significantly reducing IFN-γ secretion [89].

## 5. Use of GNP in Antiviral Immunization

GNP is a favored tool of virologists and has been used frequently in the development of the antiviral vaccination process.

### 5.1. HIV

Human immunodeficiency virus (HIV) possesses an important cluster of mannose-rich glycans in its envelope glycoprotein called gp120, which is recognized by 2G12-like antibodies. Gold nanoparticles were synthesized with a monolayer coating of self-assembled oligomannosides (similar to gp120) and were capable of binding with 2G12 [90]. GNPs attached with thiol-terminated oligosaccharides have also been used for developing HIV vaccines [91].

GNPs 2 nm in size coated with a synthetically prepared partial structure of multiple mannosidases [91,92] provide excellent binding to anti-HIV antibody-like 2G12. The third variable region (V3 peptide) of gp120 of HIV1 forms alpha helix or beta-strand conformation, which can be conjugated to GNP. This makes them more stable against any form of peptidase degradation and can also produce a high amount of specific neutralizing antibodies in rabbits [92].

Rabbits were immunized intramuscularly with 50 μg of 2 nm glyconanoparticles coated with the V3β peptide of the HIV-1 gp120 protein. Post-immunization, they produced a high titer of neutralizing antibodies against HIV1 [91,93]. Moreover, Gag p17 peptide of HIV1 conjugated with 2 nm GNP exhibited an increased proliferation of cytotoxic and helper T cells specifically against HIV, along with functional IL-1β and TNF-α cytokine production [93]. GNPs conjugated with Gp120, and gp41 HIV proteins have also been tested for vaccine development.

### 5.2. Hepatitis B

Hepatitis B virus surface antigen (HBsAg) DNA coated with GNP was injected into the epidermic cells employing a gene gun as a possible treatment measure [94]. GNPs were also used as adjuvants along with plasmid DNA encoding HBsAg DNA and injected into mice. The presence of GNPs triggers fast antibody production that leads to a quick achievement of the peak antibody titer in the animals [95].

In vitro studies in RAW 264.7 macrophages with a gold nanocage conjugated with HBsAg showed better uptake and antigen processing with IL4 secretion [96]. Recent advances have been made in the detection and diagnosis of the HBsAg by using GNP [97,98].

Virus-like particles (VLP) were produced with 10 nm GNPs conjugated with CpG oligodeoxynucleotides (ODN) and core antigen of hepatitis B. Mice immunized four times with 50 μg conjugate (i.p.) showed a 200% increase in the antibody titer as compared to GNP-free administration. CD4 helper T cells and CD8 cytotoxic T cell population expanded with the higher secretion of IL-4 and IFN-γ, along with immunostimulation of both Th1 and Th2 responses [62].

### 5.3. Hepatitis C

An interesting and effective means of delivery of hepatitis C virus DNA vaccine was proposed by a group that used plasmonic GNP activated by an electrical charge. This led to increased pore formation on the cell membrane and enhanced uptake of the DNA vaccine. The immunized mice group exhibited 100 times more gene expression as compared to the control group (without GNP use), with highly activated humoral and cell-mediated immunity being reported [99]. E2 proteins of hepatitis C were used to conjugate with GNP and immunize the mice to obtain higher igG production and proliferation of splenocytes [100].

### 5.4. Dengue

GNPs 20, 40, and 80 nm in size were used to conjugate the serotype 2-derived domain III envelope glycoprotein of dengue virus (EDIII). The conjugate administered three times subcutaneously to BALB/C mice led to the production of serotype-specific concentrated neutralizing antibodies. The size and concentration of GNPs were manipulated to affect the levels of antibodies produced in the animals. Splenocyte proliferation, helper, and cytotoxic T cell expansion and activation with the increased synthesis of IL-4 and IFN-γ were observed in mice [16]. GNP was also conjugated with small interfering RNA (siRNA) produced against the dengue virus. GNP in this conjugation helped in the enhanced stability and delivery of siRNA and elicited a better immune response [101].

### 5.5. Influenza

Much work on the development of a vaccine against influenza by using the GNP has been carried out by Gill’s group. They took a highly conserved N-terminal conserved extracellular domain of influenza virus matrix protein 2 (M2e) peptide and conjugated it with a 12 nm GNP [102,103]. They used soluble CpG and CpG ODN as their adjuvants. BALB/C mice were immunized two times with the conjugates, which led to enhanced production of IgG1 and IgG2 and better protection against a lethal dose of PR8-H1N1 infection challenge [104].

Another group has shown that even after 15 months of vaccination with GNP/M2e+ CpG conjugate, the mice retained M2e-specific neutralizing antibody production and could survive the lethal H1N1 challenge. They suggested that the vaccinated mice could effectively retain the memory B cells specific to the M2e peptide used [105].

Two intraperitoneal doses of M1 antigen of influenza virus conjugated with 15 nm GNP led to a higher titer of neutralizing antibody production along with the synthesis of IFN-γ and interleukins (ILs) 1β and 6. The activation of spleen lymphocytes and peritoneal macrophage respiration was also reported [106].

Another study proposed the use of more than one antigen against influenza in the same vaccination dose. They prepared and administered GNPs conjugated with hemagglutinin and flagellin of the H3N2 influenza virus. This vaccination process generated stronger systemic and mucosal immunity and better protected the animals from the lethal influenza challenge, compared to when a single antigen conjugate was used [107,108].

Many other vaccination processes have been developed against viral pathogens. A few of them are highlighted in Table 1.

Table 1 some of the studies focusing on the use of GNP-based nanovaccines against viral pathogens. Surface proteins from the viruses are tagged on the GNP to develop the nanovaccines. Most of the studies focus on mice models for demonstrating immune activity. The key factor of these studies is the successful development of the antibodies with neutralizing capacity [114]. Although one study reports the effectiveness of their novel GNP vaccine to be superior to the commercially available one [109], many other studies lack this important criterion to investigate. Enhanced activity by the professional antigen-presenting cells, such as dendritic cells and macrophages, is reported [110,111,112,114].

Expert opinion and future perspectives on Section 5 and Table 1: Most of the studies regarding the antiviral use of GNP-based vaccination programs focus mostly on the humoral or antibody-based immune protection by the host. Antibody titers are taken as the primary consideration to evaluate the efficacy of the novel immunization program. The cell-mediated immunity and particularly the maturation differentiation activation status of the dendritic cells, T cell subtypes, etc. (Figure 2B,C), are not studied in detail in many of these studies. Studying in depth this branch of immunity might answer various questions that are yet to be addressed.

Apart from the GNP-based influenza vaccine development (Section 5.5.), most other studies have not explored the possibility of the use of multiple protein epitopes of the pathogens conjugated to the GNP surface. Studying that aspect might expand the possibility of conferring a broad protective immunity against the pathogen. Future studies of isolating the memory cells (Figure 2D) from the vaccinated host and transferring them to naïve animals might be interesting for exploring the possibility of long-term immune protection. The comparative profiles of the routes of administration of the novel GNP-based vaccines are not disclosed or presented in most of these studies. Each study has shown either subcutaneous or intramuscular or intraperitoneal mode of delivery. The question remains on the possibility of a better immune response by the host if the vaccine is delivered through other delivery routes.

## 6. Use of GNP in Antibacterial Immunization

GNP is used for designing and delivering antigens for immunization in a number of bacterial infections. In some cases, it also acts as an adjuvant. Antigenic fragments from bacterial sources are tagged along with the GNP to stimulate the immune response generated against them.

A vaccine against the N terminal domain of the flagellin subunit of *Pseudomonas aeruginosa* along with GNP and Freund’s adjuvant induces a better IgG response [23]. Two antigens from *Francisella tularensis* were isolated and conjugated with 15 nm GNP to immunize the animals and obtain antitularemia sera rich in neutralizing antibodies [115]. In another study, 15 nm GNP was used as the adjuvant for the first time during the preparation of antibodies against the surface of the antigens of *Yersinia pseudotuberculosis* [116].

Another group studied the efficacy of the antibodies raised against F1 antigens of *Y. pestis* after coating it on 15 nm GNP. It helped elicit igG2a levels, interferon gamma, and Th1 cell activation [21]. Similarly, synthesized surface antigens of *Salmonella typhimurium* were also coated on GNPs and were reported to have better immunogenic properties in the clearance of the bacteria [117]. Non-immunoactive mono- and disaccharides derived from capsular polysaccharides of *Neisseria meningitidis* were coated on GNP and reported to have better T cell activity, MHCII presentation, and immune properties [118].

Several other uses of GNPs in the immunization process against bacterial infection are discussed in Table 2.

Table 2 some of the studies focusing on the use of GNP-based nanovaccines against bacterial pathogens. Successful production of neutralizing antibodies is the key to combating pathogens. These studies demonstrate that the GNP-based nanovaccine formulation can successfully combat these bacteria. The use of adjuvants makes the nanoformulations perform better [26,121,122,123]. The activation of the T cell subsets in most of these studies is an indicator of dual combating potential by means of cell-mediated and humoral-mediated immunity. It is interesting to note that there is only a minor difference in the immune response based on the route of administration. Some studies use more than one route of vaccine delivery [24,119,121,122,123]. The cytokine response is inclined to a pro-inflammatory or Th1 immune response [19,121,122,123], creating ideal conditions for the clearing of the pathogens.

Expert opinion and future perspectives on Section 6 and Table 2: It is more common to take vaccines against the virus than bacterial pathogens. Although most of the studies here show promising results for the use of vaccines against multiple types of bacteria, it is not yet clear how the host benefits from the antibacterial vaccines in comparison to the available antibiotics. It would be relevant to compare the controlled drug release via GNPs in the infectious site with that established in many GNP-based cancer vaccination programs. Many FDA-approved drugs and antibiotics are used to conjugate with GNP for various treatment possibilities. Some of these antibiotics are Ciprofloxacin [9], Lincomycin [10], Vancomycin [11], Ampicillin [12], Cefaclor [13], Rifampicin [14], and Kanamycin [15].

Most of the animal studies discussed in Section 6 and Table 2 did not follow up with the host for a considerable period after the vaccination. Thus, the duration and the strength of the immune protection conferred by these vaccines remain unclear. Gold nanoparticles are reported to have adjuvant properties for boosting the immune system. It would be interesting to study what proportion of the host immune protection is derived from the GNP alone as compared to the combination of other adjuvants used in these studies. In Table 2, most of the GNPs used range from 15 to 25 nm in size. As it is already well known that the size of the GNP influences the immune response, it would be useful to study the comparative account of the use of differently sized GNP in these studies.

## 7. Use of GNP in Anti-Parasitic Immunization

Some parasitic infections are being studied where GNP plays a critical role in generating the immune response in the host to combat the infections. The following table describes a few of them.

Table 3 some of the studies focusing on the use of GNP-based nanovaccines against parasitic pathogens. The activation of both MHC I and II was reported, along with both the CD4T and CD8T response [124]. These are the keys to fighting against pathogens. The high titers of specific antibodies [31,32] with better host responses against the pathogens were observed.

Expert opinion and future perspectives on Section 7 and Table 3: The studies aiming at GNP-based vaccination in parasitic diseases are relatively few, and there are various scopes to improve and explore. Plasmodium falciparum is one key pathogen mostly studied by various groups because of its wide prevalence and potential to cause deaths worldwide.

In the previous three sections of this review, we discussed the successful laboratory implementation of GNP-based nanovaccines. It would be interesting to explore and carry out a comparative study of treatment with the antibody synthesized by using GNP-conjugated antigens side-by-side with the GNP nanovaccine formulations to determine which treatment works best in a particular infection. GNPs conjugated with the antigens, haptens, and adjuvants (such as Freund’s or alum) of various pathogens have been used to obtain the antibodies. Some of these pathogens are dengue viruses [16], foot-and-mouth disease [17,18], influenza [19], *Escherichia coli* [20], *Yersinia* [21], tetanus toxoid [22], *Pseudomonas aeruginosa* flagellin [23], *Listeria monocytogenes* [24], *Streptococcus pneumoniae* [25], *Burkholderia mallei* [26,27], *Neisseria meningitides* [28], tuberculin [29,30], and malaria plasmodium surface proteins [31,32].

## 8. Limitations of the GNP

The use of the GNP shows some promising results in nanovaccine development technology. Still, it is not free from limitations. GNP, being a non-biodegradable agent, can easily be accumulated in vivo, which eventually might lead to certain side effects. Such non-porous and non-biodegradable properties might impair GNPs’ impact on the encapsulation and the timed or targeted release.

Biosafety is a major concern when using any nanomaterial. A research group has shown that encapsulated GNP conjugated with fluorescein isothiocyanate (FITC) suppresses reactive oxygen species and cytokine secretion from the macrophages [125]. GNP size-dependent toxicity is reported with smaller diameters (1–2 nm) that can be internalized by cells and organelles such as nuclei and mitochondria, leading to the induction of irreversible cellular damage [126,127]. GNPs more than 15 nm in diameter are mostly localized to the cytoplasm without being uptaken by the organelles [127]. Meanwhile, 20 nm GNPs cause oxidative stress, activate the autophagic pathway, and finally lead to genomic instability [128]. Other groups have shown lysosome impairment [129] and higher mitochondria [130], endoplasmic reticulum, and Golgi apparatus [131] accumulation of GNPs within the cell. Thus, we can see reports of the disruption of cellular metabolism due to the accumulation of GNPs in cells and their organelles.

The low penetration depth of GNPs due to the photothermal effect is a limiting factor to release the drugs or vaccinating agents into the depth required, leading to lessened immunoregulatory activities [132]. Surface modifications on the GNPs can lead to the alteration of the histocompatibility and pharmacokinetic parameters [133]. Thus, each variant of the GNP must be characterized individually before being used in therapeutic or clinical settings.

Moreover, there is still a lack of in-depth understanding about GNPs’ influence upon interaction with different cell types, especially after the modifications. Although reports suggest the formation of reactive oxygen species (ROS), oxidative stress and cell cycle impacts with induced DNA damage are also possible biological cellular responses [134,135,136,137]. GNPs coated with 1.4 nm triphenyl monosulfonate induced oxidative stress within the cells, with mitochondrial potential loss leading to necrosis [138]. The endogenous redox capacity of the cells was also impaired by GNPs by depleting the naturally available antioxidants in the cells [138].

Positively charged GNPs are reported to have a more toxic effect due to their propensity toward negatively charged DNA and cell membranes. However, both positively and negatively charged GNPs, and not neutrally charged, have been reported to have harmful impacts leading to mitochondrial stress [139,140].

## 9. Discussion and Future Perspectives

In this review, we discussed the promising potential of the gold nanoparticle for prophylactic and therapeutic purposes. Advancements in the field of nanotechnology coupled with vaccine research have paved the way for successful nanovaccine development for combating many deadly infections. Being relatively safe to administer in humans, GNPs are widely used in the development of vaccines for many diseases, ranging from cancer to infections. Some of the vaccines developed against different cancer forms are in clinical trials and show promising outcomes. The summarized advantages and limitations of the GNP in its use in the vaccine development process are tabulated in Box 1.

Box 1Advantages and limitations of the use of GNPs in the vaccine development process.
**Advantages**
BiocompatibleEasy synthesis processSize- and shape-dependent varied immune responseColloidal stabilityOptical propertiesEfficiency in molecule loading on the surfaceSurface functionalization flexibility and multi functionalization propertyCan be designed for targeted delivery and controlled release of drugsPhotothermal conversion potentialInherent adjuvant potentialUsage in imaging techniquesHigh binding affinity with wide range of moleculesHigher surface area to volume ratioLarge surface energy and charge

**Disadvantages/Limitations**
Non-biodegradableNon-porousLimited penetration depthAltered biodistribution profile upon surface modificationSurface functionalization-mediated toxicity and pharmacokinetics issuesLimited knowledge of impact on multiple cell typesClearance by macrophage phagocytosis system and renal pathwayAccumulation in cellular organelles such as mitochondria, lysosomes, etc., hampering normal cellular metabolism and ROS production


The properties of gold nanoparticles and the ease of their usage and functionalization make them attractive to researchers. Attaching the isotopes or fluorochrome tags or optical probes with gold nanoparticles and targeting them to specific cells by attaching the antibodies or targeting ligands remarkably helped the advancement of optical imaging techniques, as well. Another success of using GNPs has been achieved in obtaining the antibodies for immunological identification of different pathogens in biosensor or microscopic methods. GNP antigen-conjugated vaccines are reported to protect animals from a lethal dose of virulent challenge with a 100% survival rate [140].

There are certain concerns regarding the role of GNPs in the inhibition of Th1 response, which is crucial in combating many pathogenic infections. Only one study reported the enhancement of Th1 as well as Th17 immune response; the authors used the Listeria antigen along with the combination of Advax and 25 nm GNP adjuvants [24]. Most of the other studies highlight the increase in the Th2 response post-GNP administration. However, this shortcoming inactivation of the proinflammation by the GNPs themselves is overcome mostly by the antigens or drugs they are carrying or the cells they are targeting via the attached ligands.

To improve prospects regarding the use of GNPs in vaccine development and in clinical settings, there is a pressing need to address certain issues. Firstly, there must be a large-scale production setup for GNPs with a high level of consistency. As we have addressed before, multiple variable factors such as charge, size, and shape all impact the cells in different ways. Thus, we need to be clear and cautious about each change that we are implementing. We have also noticed that most labs are working with GNP sizes ranging from 15 to 50 nm. Moreover, nanoshell-structured GNPs are most often used rather than other shapes such as nanocage, nanorods, nanocubes, etc. Therefore, we lack knowledge of the GNPs sized or shaped differently than as mentioned.

Secondly, it is important to characterize with better clarity the GNPs’ impact upon interaction with immune cells. We believe a detailed investigation of immune cells in the presence of functionalized or empty GNPs will be helpful for answering various questions. Third, the biodistribution of the GNP must be evaluated in further detail, with special emphasis on the off-target cells and organs. Most studies highlight only the organs or cells or the disease pathogen and do not consider the possible accumulation of the nanomaterial in other organs. This leads to our fourth concern: nanotoxicology. As already discussed, the GNP is a non-biodegradable substance; thus, there may be a need to develop a synthesis or tagging method that can make it less toxic or better suited for clearance.

Fifth, the protein coating formed outside the GNP surface upon in vivo introduction (also known as bio-corona) is a problematic factor for the efficacy of the conjugated antigens on its surface. This bio-corona formation blocks the coating antigens and materials from interacting with in vivo cellular physiology. Sixth, as there is no specific dosage yet, the problem remains of replicating the success observed in animal experiments in early clinical trials. The standardization and normalization of the dose must be established.

Seventh, a key important issue that needs to be addressed is how to fashion the GNP so that it can evade clearance by MPS or by renal excretion before its intended action. Studies have suggested coating GNPs with PEG, polyvinyl alcohol, poly (acrylic acid), or biomolecules such as glutathione or albumin to prevent bio-corona formation and MPS-based clearance and provide stability with relative less off-target toxicity [64,141,142]. Eighth, as the use of gold always bears a cost, logistical concerns regarding the manufacture and distribution of such vaccines across a wide range of populations must be considered.

Future work should address the impact of the combination strategies with GNP-based delivery along with photothermal therapy. It would also be interesting to explore the optical properties of GNPs in combination with thermal therapy in inflammatory responses. We expect that although some of the GNP-based nanoformulations discussed in this review might be translated into clinical settings, it is vital to address the multiple challenges associated with GNPs. Therefore, of paramount importance are the balanced testing and validation of their safety before establishing them in biomedical applications.

## Figures and Tables

**Figure 1 vaccines-10-00505-f001:**
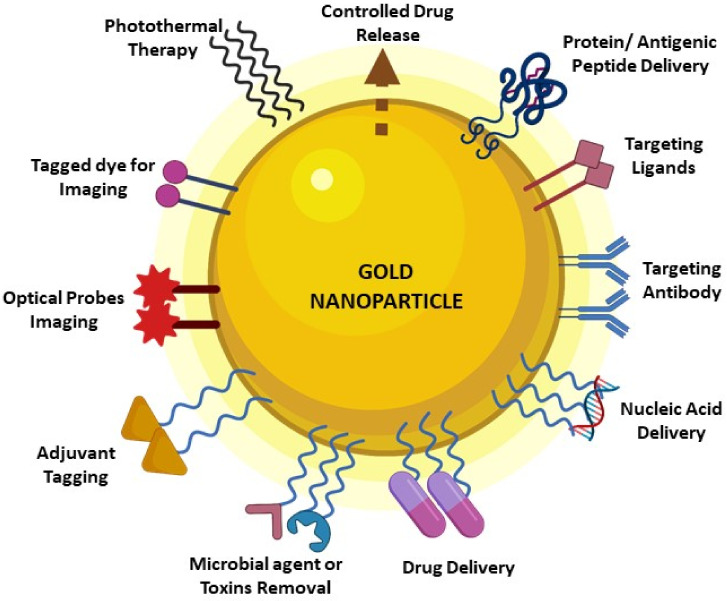
Schematic representation of the gold nanoparticle and its possible uses in the biomedical field. The gold nanoparticle can be tagged with one or more things depending on its intended use, such as the delivery of nucleic acids or protein fragments or the delivery of drugs and their controlled release. Targeting of its contents to the specific cells of the body is performed by using antibody-tagged GNPs or by the use of ligands attached to them targeting specific receptors of the body. Imaging techniques have been immensely developed by the use of GNP-tagged dye optical probes.

**Figure 2 vaccines-10-00505-f002:**
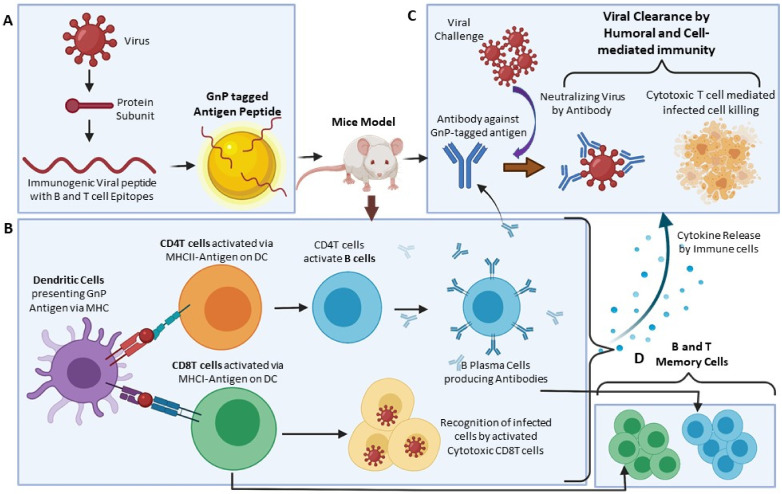
A schematic representation of the use of GNPs in developing nanovaccines against a virus. (**A**) The protein subunit from the virus is isolated to determine the peptide sequence, which is both immunogenic for the host and conserved across multiple strains of the virus. The peptide is tagged with GNPs to create the novel nanovaccine and tested on the mice model. (**B**) The immune cells of the mice are triggered as the dendritic cells start presenting the peptides to the CD4 helper T cells and the CD8 cytotoxic T cells. The clonal expansion of the activated helper T cells and subsequent activation of the B cells into the plasma cells lead to the production of the antibodies specific to the peptide used for nanovaccine production. The cytotoxic T cells can recognize and deploy themselves in the killing of the infected cells. (**C**) The cytokines produced during the process of immune regulation of the nanovaccine produce a chemical milieu where the immune cells can favorably fight against the pathogens and shape the Th1 or Th2 immune response depending on the inflammation status. The antibodies can recognize the peptide sequence present in the whole virus and neutralize them effectively. (**D**) The B and T memory cells formed during this vaccination process can hold the information of the peptide used during the process and live long after. They are equipped to start an immediate immune response against any future challenge of the same virus and thus can eliminate them before they can cause major harm to the host.

**Table 1 vaccines-10-00505-t001:** Use of GNP based nanovaccine against viral pathogens.

SN	Antigen Conjugated with AuNP	GNP/Adjuvant	Immunization Mechanism	Immune Response	Ref.
1	Surface antigens spike glycoprotein of avian coronavirus	Virus-like particles (VLP) by incubating the antigen with 100 nm AuNPs	Dose: Single, 10 μg Mode: Intramuscularly Animals: BALB/C mice and specific pathogen-free chickens	Showed increased antigen delivery to lymphoid organs. An enhanced response of spleen T cells. Higher antibody titers. A reduction in symptoms of infection.(Comparative study with a commercial vaccine also showed that the AuNP conjugate provided better protection against the virus.)	[109]
2	Surface antigens gastroenteritis virus	Conjugated with 15 nm AuNPs	Guinea pigs twice subcutaneously with 125 μg, mice once intraperitoneally with 70 μg, and rabbits three times subcutaneously with 220 μg	Increased the level of IL-6, IFN-γ, IL-1β in the blood plasma. Higher respiratory activity of peritoneal macrophages and spleen lymphocytes. Activation of humoral immunity; increase in the number of follicles in the spleen.	[84,110]
3	Glycoprotein antigen of respiratory syncytial virus	Nanorods	Human cell treatment in vitro	Human dendritic cells induced an immune activation (proliferation and expansion) of primary T cells.	[111]
4	Glycoprotein isolated from fixed rabies virus, strain Moscow 3253	Conjugated with 15 nm AuNPs	Animal: Mice Mode: Intraperitoneally Dose: 25 μg in four booster doses, 50 μg was used	Develop highly specific neutralizing antibodies against the virus.	[112]
5	Surface glycoprotein (gB) of human cytomegalovirus (CMV, a herpes virus)	Conjugated with AuNP	In vitro	Viral replication blocked. Virus-induced cytopathogenic effects blocked. Virus spread in cell culture decreased without generating cytotoxicity. Cells gained resistance to CMV infection post-treatment.	[113]
6	West Nile fever virus	Multiple sizes and shapes of AuNPs used: 20 and 40 nm nanospheres, 40 × 20 nm nanorods, and 40 × 40 × 40 nm nanocubes	Animal: Mice Mode: Intraperitoneally Dose: 100 μg No. of doses: 2	40 nm nanospheres induced the highest level of specific antibodies. The dendritic cells and macrophages absorbed larger numbers of nanorods. IL-1β and IL-18 synthesis increased while using nanorods, while nanospheres and nanocubes resulted in higher synthesis of TNFα, IL6, IL12, and granulocyte-macrophage colony-stimulating factor.	[72]
7	Capsid (Cap) protein from pathogenic porcine circovirus	Conjugated with 23 nm GNPs	In vitro and mice immunized twice subcutaneously	Increase in Cap protein phagocytosis. High production of virus-neutralizing antibodies.(Similar results were obtained with classical swine fever virus antigen.)	[114]

**Table 2 vaccines-10-00505-t002:** Use of GNP based nanovaccine against bacterial pathogens.

SN	Antigen Conjugated with AuNP	GNP/Adjuvant	Immunization Mechanism	Immune Response	Ref.
1	Listeriolysin O peptide (LLO91-99) from *Listeria monocytogenes*	Conjugated with AuNP	A single intravenous or intraperitoneal immunization of mice	Increase in the number of splenic CD4+ and CD8+ T cells, NK cells, and CD8α+ dendritic cells specific T cell response. An increase in the synthesis of the cytokines IL-12, TNF-α, IFN-γ, and MCP-1. Newborn mice born to vaccinated females were healthy and bacteria-free.	[95,119]
2	A synthetic tetrasaccharide epitope, similar to the capsular polysaccharide of *Streptococcus pneumoniae* type14	Conjugated with 2 nm AuNP + T helper peptide	Animal: Mice Dose: 3 μg Mode: Intradermal No. of doses: 1	Specific high-titer IgG. Increase in the level of the cytokines IL-2, IL-4, IL-5, IL-17, and IFN-γ. Increased phagocytosis of type 14 bacteria stimulated by antisaccharide antibodies.	[25,120]
3	Bacterial vesicles of the outer membrane of *Escherichia coli*	Conjugated with 30 nm AuNPs	Injected in mice three times subcutaneously	Rapid maturation and activation of dendritic cells in the lymph nodes. Increase in higher avidity antibodies. Enhancement of IFN-γ and IL-17, indicating strong Th1 and Th17 cellular responses.	[19]
4	Tetanus toxoid *Clostridium tetani*	Conjugated with 25 nm AuNPs + plant adjuvants (saponins) from *Quillaja saponaria* (79) and *Asparagus racemosus* (80)	Subcutaneous injection, or transmucosal delivery	Oral administration highly enhanced mucosal immune response in the presence of plant adjuvants.	[121,122,123]
5	*Burkholderia mallei* recombinant protein: Hc fragment of tetanus toxin, hemolysin (produced by both B. mallei and *B. pseudomallei)*, and flagellin (produced by *B. pseudomallei)*	15 nm AuNP functionalized with purified LPS from a nonvirulent *B. thailandensis* strain	BALB/C mice, intranasal, 3 different dose concentrations	Generated significantly higher antibody titers compared with LPS alone. Improved protection against a lethal inhalation challenge of B. mallei in the murine model of infection.	[26]
6	7.5 μg of tuberculin (mixture of the surface antigens of various types of mycobacteria)	Conjugated with 15 nm AuNPs	Rabbits, four times intramuscularly	High antibody production against multiple types of mycobacteria.	[29,30]
7	Specific immunogenic antigens LomW and EscC from enterohemorrhagic strain *E. coli* O157: H7	Conjugated with AuNP	Mice, three times subcutaneously, 2-week intervals	Higher-titer IgG and IgA. Serum IgG titer increase correlates with the decrease in the intestinal colonization of *E. coli.* Reduced the adhesion of *E. coli* O157: H7 and two different *E. coli* pathotypes to humans. Bactericidal properties of intestinal epithelial cells specific to antigen generated.	[20]

**Table 3 vaccines-10-00505-t003:** Use of GNP based nanovaccine against parasitic pathogens.

SN	Antigen	AuNP/Adjuvant	Immunization Mechanism	Immune Response	Ref.
1	Recombinant protein from rSm2 *Schistosoma mansoni*	Gold nanorods conjugated	Mice immunization intraperitoneally with 2 μg dose	Th1 immunological response. Higher production of IFN-γ, mostly by CD4+ and CD8+ T cells. Activated dendritic cells (in vitro). Increase in the expression of MHCI and MHCII and the synthesis of IL-1β.	[124]
2	Surface protein Pfs25 from the *P. falciparum*	Attached to various AuNPs, including nanospheres, nanostars, nanocages, and nanoprisms	Mice were immunized with the resulting conjugates. Dose: 10 μg, three times, intramuscularly	High-titer antibodies. The highest titers were obtained with gold nanospheres and nanostars. The antibodies blocked the transmission of parasites to mosquitoes in membrane-feeding assays.	[32]
3	C-terminal 19 kDa fragment of merozoite surface protein 1 from the malaria pathogen *Plasmodium falciparum*	17 nm AuNP conjugated + adjuvant Alhydrogel^®^	Mice were immunized three times subcutaneously at a dose of 25 μg	Antibodies produced against the weakly immunogenic peptides. It blocked the invasion of *P. falciparum* in an in vitro assay.	[31]

## Data Availability

Upon request we can, within a reasonable timeframe, provide data described in this manuscript.

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
