# Peer review of "Efficacy and Immune Response Elicited by Gold Nanoparticle- Based Nanovaccines against Infectious Diseases"

_vaccines, 2022, doi:10.3390/vaccines10040505_

Round 1

Reviewer 1 Report

The manuscript mainly highlighted the role of Gold nanoparticles in vaccine development for infectious diseases. The manuscript contains weak arguments and very less valuable information. In the current form, the manuscript cannot be considered for publication in this journal. The author needs to revise the manuscript extensively, include valuable information and critically evaluate characteristic features of Gold NPs for vaccine development. Furthermore, some suggestions are given below for future submission;

  1. The authors are claiming that Gnp is not toxic in the second line of abstract. This is absolutely debatable. The authors shall explain it later in the manuscript that how Gold has no toxicity or revise the sentence. The authors themselves claiming in section 2 that Gnp has less toxicity.
  2. The whole abstract should be revised, the authors should focus more on the benefits of Gnp as a delivery vehicle, then its effect on DCs etc. In abstract, the authors shall briefly highlight its surface chemistry, composition, ease in modification, optical properties and other features, which make it unique for Vaccine development.
  3. The authors shall use abbreviation at first usage and use the abbreviated term in the rest of the manuscript. For example: Second line of the Abstract- Gold Nanoparticles (Gnp).
  4. The authors are encouraged to expand the introduction section. It is recommended that the authors consider to read extensive literature related to Gold Nanoparticles.
  5. There are no legends to all the tables in the manuscript.
  6. The grammar and English fluency is very weak. The authors need to revise the English and write the manuscript in scientific language.
  7. The section 2 is written very rough. There is no proper explanation of the characteristic features of Gnp in detail. The authors need to explain each characteristic in detail by highlighting some important studies conducted by leading researchers. They should also highlight the outcome of the studies.
  8. In section 3, The abbreviations “BMDC, GMCSF” and many others are used. Please write the full names of these abbreviations before writing short form. Careful consideration should be given to all the abbreviations used in the manuscript. There are many mistakes.
  9. There are many mistakes in references. The references do not represent the study included in this manuscript. The authors shall cite the exact reference for each statement in the article. For example, Reference 23 is not the exact reference of the study included in this manuscript. The author should check all the references whether the exact reference is cited or not.
  10. The section 5 and 6 needs much improvement, and the authors must include more information.
  11. In section 8, the author claims that GnP vaccines undergoing many clinical trials for cancer and infectious diseases. The authors should include a table of those clinical trials. Over all, the Discussion part is weak, and needs extensive and critical arguments.
  12. What is the conclusion of authors from their own manuscript?
  13. References are not unified. The authors need to unify all the references.

Author Response

Please find the reply in the document.

Reviewer 2 Report

Review of Manuscript “Efficacy and immune response elicited by gold-nanoparticle based nanovaccines against infectious diseases ” (Review) by Sengupta et al..

The review by Sengupta et al. addresses a potentially very interesting topic, the use of Gold nanoparticles (GnP) as carriers for vaccination against a variety of infectious diseases.

However, the manuscript suffers from a very poor quality in terms of content and language. In view of the large number of linguistic errors, it would go beyond the scope of this review to go into all of them in detail. However, in a large portion of the text one finds inappropriate verb forms and sentence fragments. An absolute minimum requirement for resubmission would be post-processing by a native English speaker in order to make the content really understandable.

In terms of content, it is not much more than a mere listing of references without any critical scientific evaluation and classification. The presentation of the data in the tables 1 and 2 in no way meets the requirements for a review in a peer-reviewed journal. In addition, figures 1 and 2 are not referred to in any way in the running text. Figure 2 merely represents textbook knowledge at a very basic level.

In summary, in my opinion the manuscript is not suitable for publication in its present form.

Author Response

Please find the reply in the document.

Reviewer 3 Report

Generally informative review, but the style of writing (the tone) might need to be carefully readjusted. Minor comments are listed below:

  • It is surely important to highlight the strength of the materials review, but overselling it is not going to look right. For example, the author claimed that gold is one of the most successful nanoparticles used in vaccine development, clearly not supported by real-world evidence. Lipid-based nanoparticles are acquiring much more clinical success that has already been clinically approved and used worldwide. Overselling is a general issue throughout the manuscript that needs to be addressed. Being objective is key to any review article.
  • Following 1, the rationale for using gold over other materials for various applications was not well-explained/justified. Note that having some paper in the literature reported is not a justification. Help the reader by summarizing what is important about using gold nanoparticles is more important than showing what gold NP can do in Figure 1 because any nanoparticle can do more or less that.
  • Following 1-2, in that regard, there is a lack of perspectives out of the tables summarized that is unique to the use of gold nanoparticles, especially regarding the immune responses. Right now, it is just a random collection. Help the reader by summarizing the direction, and summarized finding out of the review will make the article great.
  • Lack of summaries and discussion on the clinical translation of GnP. What are good, bad, and limitations? Critically reviewing the limitation will pave the way forward for GnP.

Author Response

Please find the reply in the document.

Round 2

Reviewer 1 Report

The manuscript is revised in terms of language and some valuable information. Important literature has been discussed. Overall, a good attempt has been made by the authors. Some comments are given below.

  1. There is still a lack of appropriate figures which describe the content of the manuscript. For example, the authors describe GNPs as a potential candidate for vaccine development, so they must include a figure describing how it can become a better vaccine candidate in a schematic way in a diagram.
  2. Line 27, use either Gold nanoparticles or GNPs as keyword.
  3. Either use Figure or Fig in the content of the manuscript (Line 61 and line 72). Please check all similar mistakes.
  4. The legend is always on top of the table. Please pay attention to it.
  5. Please remove * from tables.
  6. Previously I asked you to revise the references. The references need manual correction.
  7. The author still needs to work on figures.
  8. According to their claim of GNPs as a potential vaccine candidate, then a table must be incorporated regarding clinical trials on GNPs as a vaccine candidate if any?
  9. A point-by-point response is necessary. 

Reviewer 2 Report

Review of revised version of manuscript “Efficacy and immune response elicited by gold-nanoparticle based nanovaccines against infectious diseases ” (Review) by Sengupta et al..

The authors have addressed most of the concerns in my review of the original version of the manuscript in a satisfactory fashion.

Especially the English language has improved significantly, making the review much easier to read and understand.

Although the presentation of most of the relevant references in form of two tables still has some limitations, the scientific evaluation of these references has also improved greatly through the extended changes in the novel section 9 (section 8 in the original manuscript).

Furthermore, I concede that the basic schemes in the figures may be helpful for readers not quite so familiar with immunoregulatory pathways.